# Deciphering White Adipose Tissue Heterogeneity

**DOI:** 10.3390/biology8020023

**Published:** 2019-04-11

**Authors:** Quyen Luong, Jun Huang, Kevin Y. Lee

**Affiliations:** 1Department of Biomedical Sciences, Heritage College of Osteopathic Medicine, Ohio University, Athens, OH 45701, USA; ql255013@ohio.edu (Q.L.); jh611517@ohio.edu (J.H.); 2The Diabetes Institute, Ohio University, Athens, OH 45701, USA

**Keywords:** development, obesity, adipose tissue, metabolic syndrome

## Abstract

Adipose tissue not only stores energy, but also controls metabolism through secretion of hormones, cytokines, proteins, and microRNAs that affect the function of cells and tissues throughout the body. Adipose tissue is organized into discrete depots throughout the body, and these depots are differentially associated with insulin resistance and increased risk of metabolic disease. In addition to energy-dissipating brown and beige adipocytes, recent lineage tracing studies have demonstrated that individual adipose depots are composed of white adipocytes that are derived from distinct precursor populations, giving rise to distinct subpopulations of energy-storing white adipocytes. In this review, we discuss this developmental and functional heterogeneity of white adipocytes both between and within adipose depots. In particular, we will highlight findings from our recent manuscript in which we find and characterize three major subtypes of white adipocytes. We will discuss these data relating to the differences between subcutaneous and visceral white adipose tissue and in relationship to previous work deciphering adipocyte heterogeneity within adipose tissue depots. Finally, we will discuss the possible implications of adipocyte heterogeneity may have for the understanding of lipodystrophies.

## 1. Introduction

In 2015–2016, the Centers for Disease Control and Prevention (CDC) found that ~40% of adults in the United States of America, representing 93 million individuals, are obese (https://www.cdc.gov/nchs/data/databriefs/db288.pdf). This obesity pandemic and the associated co-morbidities of obesity including cardiovascular diseases, certain types of cancer, and type 2 diabetes have stimulated great interest in the understanding of adipose tissue. White adipose tissue (WAT) is the primary site for energy storage, and plays protective roles in thermal insulation and protection from mechanical stress [1]. WAT not only stores energy, but also controls metabolism through secretion of hormones, cytokines, proteins, specific lipid species, and microRNAs that affect the function of cells and tissues throughout the body [2,3,4,5,6,7,8]. The adipose organ consists of energy storing white adipose tissues (WAT), thermogenic brown adipose tissues (BAT), and thermogenic BAT-like adipocytes (brite/beige) interspersed within WAT. WAT adipocytes are histologically characterized by the appearance of large unilocular lipid droplet, low number of mitochondria, and small cytoplasmic space. BAT adipocytes are known for multiple small lipid droplets and high density of mitochondria. BAT and brite adipocytes have been the focus of much attention in recent studies, as the ability of BAT to dissipate energy through mitochondrial uncoupling could potentially burn excess calories and combat the epidemic of obesity [9]. Furthermore, recent studies have suggested that developmental and functional differences in adipocytes within distinct adipose tissue depots that influence the overall behavior of these depots [10,11,12]. 

In our recent study, we have used a combination of in vitro clonal cell analysis and lineage tracing in vivo to investigate heterogeneity of white adipocyte subpopulations. In this study, we identified at least three distinct subpopulations of white preadipocytes that are characterized by unique gene expression profiles and high expression of three different marker genes: Wilms tumor 1 (Wt1), transgelin (Tagln), and myxovirus 1 (Mx1), termed Types 1–3, respectively. These preadipocyte subpopulations give rise to adipocytes with unique metabolic properties and differentially respond to exogenous stimuli, including inflammatory cytokines, growth hormone, and insulin. In vivo, these three preadipocyte markers define three independent white adipocyte subpopulations that differentially contribute to the individual white adipose tissue depots [10]. Throughout this review, we will highlight findings from this manuscript. We will also introduce the different white adipose tissue depots and discuss their physiological contribution. In addition, we will discuss other studies that address the heterogeneity of adipocytes within adipose tissue depots. Finally, we will discuss the possible implications of adipocyte heterogeneity may have for the understanding and treatment of lipodystrophies. 

## 2. Adipose Tissue Depots

In general, the majority of WAT is categorized as either subcutaneous (SAT) and visceral (VAT) adipose tissue, with the exception of smaller adipose depots including the dermal WAT (dWAT) and bone marrow adipose tissue (MAT) that are distinct from SAT and VAT (Figure 1). SAT is subdivided into anterior and posterior in rodents, which anatomically approximates upper and lower body subcutaneous fat in humans. Major VAT adipose depots include those that surround the heart (epicardial/pericardial) and the intraabdominal organs (mesenteric, omental, perigonadal, perirenal, and retroperitoneal) in both humans and rodents. One distinction between human and rodent VAT is that humans have detectable omental WAT (oWAT), whereas rodents have large perigonadal WAT (pgWAT). In addition to SAT and VAT, the physiological contribution of other fat depots has begun to be recognized. dWAT is comprised from the white adipocytes are associated with the dermal skin layers, and has functions in wound healing, generation of hair follicles, and thermogenesis [13,14,15]. The dWAT is developmentally distinct from SAT and is physically separated from SAT by the panniculus carnosus, a striated muscle layer that is only present in rodents and is not clearly defined in humans [16,17]. Another adipose tissue receiving recent attention is MAT [18]. There are two major subtypes of MAT. Constitutive MAT (cMAT) is concentrated in the distal skeletal, contains larger adipocytes, and is relatively devoid of active hematopoiesis. On the other hand, regulated MAT (rMAT) is found in more active sites of hematopoiesis including the spine and proximal limb bones. Adipocytes of the rMAT are more diffusely distributed and tend to increase or decrease in response to environmental or pathological factors [19,20]. MAT adipocytes populate bone marrow with increasing age [21] and resist depletion during caloric deficit states [22,23,24]. MAT may also play important role in bone metabolism, and during caloric restriction, MAT may be the major source of circulating adiponectin [25]. In clinical studies of healthy populations as well as in populations of individuals with metabolic disease, MAT has been shown to be inversely associated with bone mineral density. Furthermore, lipolysis from MAT adipocytes can provide osteoblasts with free fatty acid and thus directly affects bone turnover [26].

## 3. Metabolic Contributions of Visceral and Subcutaneous Adipose Tissues 

Although the metabolic contributions from dWAT and BMAT are just beginning to be explored, the metabolic roles of SAT and VAT have been well-studied. VAT accumulation, i.e., central obesity, is associated with insulin resistance and increased risk of metabolic disease, while peripheral obesity from excess SAT accumulation may even be protective of metabolic syndrome [27,28,29,30,31]. Transplantation studies in mice demonstrate that these differential metabolic effects of SAT and WAT are at least partly cell-autonomous, as transplantation of SAT, but not VAT, improves metabolic parameters and insulin mediated glucose uptake [32,33], and underlying these effects are both molecular and phenotypic difference between adipocytes of different depots. Depot specific differences in both expression levels and release have been found for most adipokines between VAT and SAT [34]. On the molecular level, human or rodent preadipocytes and adipocytes from different depots and cultured under identical conditions retain distinct differences in gene expression patterns, even after many generations in culture. These included marked differences in the expression of developmental genes [35,36,37,38,39]. Homeobox A5 (HoxA5), Homeobox A4 (HoxA4), Homeobox C8 (HoxC8), Glypican 4 (Gpc4), and Nuclear receptor subfamily 2 group F member 1 (Nr2f1) levels are high in visceral fat, while Homeobox A10 (Hox10), Homeobox C9 (HoxC9), T-box (Tbx15), Short stature homeobox 2 (Shox2), Engrailed 1 (En1), and Secreted frizzled-related protein 2 (Sfpr2) levels are high in subcutaneous fat [35]. Our recent study on adipose lineages confirms these results, as cell lines derived from the stromal vascular fraction (SVF) of SAT depots have higher expression of Shox2 and Tbx15 mRNA levels and lower expression of HoxC9 and HoxA4 mRNA than those derived from VAT depots [10]. 

In addition to the expression of developmental genes, preadipocytes isolated from SAT also have increased basal levels of adipogenic genes including peroxisome proliferator-activated receptor gamma (PPARγ) and CCAAT/enhancer binding proteins (C/EBP) transcription factors [38,40] and lower levels of macrophage recruitment factors including C-C Motif Chemokine Ligand 2 (CCL2) and C-C chemokine receptor type 2 (CCR2) after exposure to high fat diet [41]. These differences in gene expression may explain many of the depot-specific phenotypic differences observed in adipocytes. Preadipocytes from SAT proliferate, differentiate, and have increased lipid accumulation compared to preadipocytes isolated from VAT [37,39,42,43,44,45]. On the other hand, VAT adipocytes are also characterized by an increased susceptibility to apoptosis, reduced lipoprotein lipase activity, and a high degree of catecholamine-induced lipolysis due to increased expression levels of beta-adrenoceptors [46,47,48,49]. Interestingly, functional differences between humans and mice WAT depots are found in the ability of these depots to give rise to brite adipocytes. In mice, SAT depots express higher expression of genes, including uncoupling protein 1 (Ucp1) involved in brown fat formation compared to VAT depots, while in WAT of human subjects, the opposite is true [50]. Similarly, exercise has been shown to induce browning of SAT in mice, while exercise in humans reduces markers of brown fat in human SAT [51,52]. This divergence in gene expression in fat depots between the two species strongly indicates that caution should be applied extrapolating mouse browning gene expression studies to human physiology.

In rodent models, the inguinal depot is most often used to be representative of SAT, and the deposition of gluteofemoral SAT, which anatomically roughly corresponds with inguinal fat in rodents, is associated with a protective lipid and glucose profile and decreased cardiovascular and metabolic risk [53]. However, in addition to gluteofemoral SAT, humans also have deep SAT (dSAT) found under the Scarpa’s fascia in the abdominal area (Figure 1). The dSAT depot has increased expression of inflammatory cytokines, a more saturated fatty acid composition, and distinct molecular and morphological differences from superficial SAT [54,55,56,57]. Thus, there are distinct differences between superficial SAT and dSAT, and dSAT shares numerous similarities with VAT [58].

## 4. Preadipocytes Have Numerous Developmental Sources 

The molecular characterization of the preadipocyte lineages in every individual depot has made an enormous contribution in understanding adipocyte biology and the heterogeneity of adipocytes. Numerous markers, including preadipocyte factor 1 (Pref1) and platelet-derived growth factor receptor beta (Pdgfrβ), have been shown to be highly enriched in preadipocytes prior to adipogenic differentiation [59,60]. A key advance in the field was fluorescence-activated cell sorting (FACS) characterization of the preadipocyte lineage. In this study, Rodeheffer et al. utilized FACS analysis to define the preadipocyte lineage as cells that are negative for markers of other committed lineages, i.e., endothelial cells (CD31), macrophages (CD45), and erythrocytes (Ter119), and are positive for the stem cell markers CD29, CD34, Sca1, and CD24. Transplantation of these precursor cells was able to generate WAT in the A-Zip lipodystrophic mouse model. Further follow-up studies determined that adipogenic CD24- cells represented a more committed preadipocyte lineage downstream of CD24+ precursors [61,62]. 

Interestingly, lineage tracing analyses indicate that preadipocytes, even those from a single adipose tissue depot, are derived from numerous developmental sources. Epithelial cells from the mesothelium have been shown to be able to give rise to adipocytes both in vitro and in vivo within VAT depots [10,63,64]. We found that these cells, which we have termed Type 1 adipocytes, have dramatically different gene expression compared to other adipocyte subpopulations with ~100-fold higher expression of many genes including—Wilms tumor 1 (Wt1), leucine rich repeat neuronal 4 (Lrrn4), and uroplakin 3b (Upk3b). In addition, we find that this subpopulation of adipocytes has reduced triglyceride accumulation and is characterized by a highly glycolytic metabolism. Interestingly, we found that HoxA5 is highly expressed in Type 1 preadipocytes, suggesting that its higher expression in the VAT is due to the presence of Type 1 adipocytes within these depots. 

Adipocyte growth has long been known to be associated with the underlying vasculature foreshadowing the more recent findings that at least some preadipocytes are found associated with the vasculature [65]. Utilizing multiple lineage tracing mouse models, Tang et al. were the first to demonstrate that a subset of preadipocytes could be found within the mural cell compartment of the adipose tissue vasculature [66]. Furthermore, the mural cell-derived preadipocytes marked by smooth muscle actin (SMA)-Cre were suggested to be important in the homeostasis of adipose tissue later in life [67]. Similarly, a vascular smooth muscle origin for a subpopulation of brite, but not brown adipocytes has also been reported, supporting the significance of this lineage in adipose biology [68]. Additionally, our adipose subpopulation marked by transgelin (Tagln) or SM22-Cre/ROSA26^mT/mG^ a gene that is highly expressed in vascular, smooth muscle, and pericytes were found in all SAT and VAT depots [10,69]. 

In addition to the mural cell-derived adipocytes, the endothelial cells of the vasculature have also been suggested to give rise to adipocytes. Progenitor cells labeled by the endothelial-specific VE cadherin-cre, give rise to both brown and white adipocytes. Similarly, Zfp423, a preadipocyte marker, is found in both mural and vascular endothelial precursors, and further supports the idea of the vascular endothelium as a source of preadipocytes [70,71,72]. However, other studies fail to show the presence of endothelial-cell derived adipocytes in either normal and high-fat-diet conditions [62]. Intriguingly, an unconventional source of preadipocytes, from bone marrow-derived cells of the hematopoietic lineage, may also contribute to the formation of other adipose tissues in addition to MAT [73,74]. The number of bone marrow-derived adipocytes accumulates with age, are found more in VAT than SAT, and are more prevalent in female than male mice [73,75]. Notably, increase in bone marrow-derived adipocytes only in female visceral fat is negatively regulated by estrogen and estrogen receptor, suggesting that this depot may contribute to VAT deposition in post-menopausal women [76]. Finally, the contribution of this subpopulation in human VAT has also been studied. In adipose tissue obtained from human recipients of hematopoietic stem cell transplantation, the hematopoietic lineage makes a sizable contribution to adipose tissue, as up to 35% of visceral adipocytes were derived from the transplantation donor [77]. 

Various potential sources of cells comprising the pool of preadipocytes reflect the diverse embryonic origin of adipose tissues. Despite the traditional thinking that all adipose tissues were derived from mesodermal tissues, subsets of adipocytes around head and neck structures have been shown to be derived from neural crest cells [78,79]. Sox10-Cre/R26–YFP, a neural crest marker [80], showed YFP+/perilipin+ cells around the ear and salivary gland in mice, indicating adipocytes in these sites are of neural crest origin [79]. However, the neural crest-derived cephalic adipocytes seemed to be a transient supply to these regions. A study using Wnt1, a marker for migrating neural crest cells, combined with Cre/R26R –YFP reporter demonstrated the progressive reduction of Wnt1-Cre adipocytes and an increase in mesodermal-derived adipocytes in neck, cervical, and pectoral areas [81]. 

The majority of adipocytes have their origin traced back to mesoderm. During early development, mesoderm is transformed into paraxial, intermediate, and lateral mesoderm. Somites are a derivative of paraxial mesoderm, and a Cre line driven by promoter of the somite-specific gene, mesenchyme homeobox 1 (Meox1), marked both dorsal and anterior white adipocytes. These depots included the retroperitoneal (rWAT) and interscapular WAT, with a few adipocytes labeled in male pgWAT. Furthermore, lineage tracing analyses show that like Meox1-Cre, transcription factors Pax3 and Pax7 give rise to a portion of interscapular WAT [11,82]. Similarly, the Myf5-Cre, a downstream target gene of Pax3/7, also gives rise to interscapular WAT and rWAT [11]. These data indicate that dorsoanterior adipose tissue depots, including the subcutaneous WAT and rWAT, are derivatives of paraxial mesoderm. A similar approach was employed to study the lateral mesoderm. Using HoxB6-Cre to mark the lateral plate, and HoxB6-derived precursors make up the majority of inguinal WAT (posterior subcutaneous fat), mesenteric WAT (mWAT), and pgWAT in female animals [82]. Other lineages that give rise to mostly subcutaneous and visceral adipocytes are paired-class homeobox 1 (Prx1)-Cre and myxovirus 1 (Mx1)-Cre marking majority of subcutaneous depots, and Wilms tumor 1 (Wt1)-Cre^ERT2^ contributing only to visceral depots [10,12,64,83]. Interestingly, while Prx1-Cre lineage gives rise to the majority of ventral subcutaneous fat, Mx1-Cre lineage contributes to majority of scapular WAT and a portion of perirenal (prWAT) which has a dorsal distribution similar to Myf5-Cre, indicating that these lineages are potentially downstream of the Meox/Pax3/Pax7 and HoxB6 lineages, respectively (Figure 2). 

## 5. Functional Heterogeneity of Adipocytes

The heterogeneous lineages observed in WAT may explain the functional differences of mature adipocytes within each depot. Adipocytes, even when isolated from a single fat depot, have been shown to have variable insulin sensitivity, fatty acid uptake, lipogenesis, catecholamine-induced lipolysis, suggesting that each depot is a mixture of numerous adipocyte subtypes [84,85,86,87,88]. Indeed, a recent study describing a detailed and optimized flow cytometry protocol for sorting mature adipocytes, found that a subpopulation of β2-adrenergic receptor (ADRB2)-expressing adipocytes are associated with the patient metabolic health [89]. Previously, we demonstrated that Tbx15 expression, which is highly expressed in SAT preadipocytes and adipocytes, marks another subset of adipocytes. As in skeletal muscle fibers, we found that in this subpopulation of adipocytes, Tbx15 regulates energy metabolism by promoting a glycolysis over oxidative phosphorylation [90,91]. Interestingly, we found subcutaneous, but not visceral Type 2 adipocytes, derived from Tagln-Cre lineage, showed high expression of Tbx15. These data suggest the presence of both Tbx15^Lo^ and Tbx15^HI^ Type 2 adipocytes. 

The epicardial fat is directly connected to the myocardium and is separated from the pericardial fat by the visceral pericardium, although many studies loosely use the term “pericardial fat” to describe both of these VAT depots [92,93]. Intriguingly, pericardial fat is not depleted during starvation, and is characterized by adipokine secretions, rapid free fatty acid uptake, high lipolytic rate, and low rates of glucose utilization [94,95]. Because pericardial fat may provide fuel for the heart and is associated with inflammation and atherosclerosis [96,97,98], understanding the function of this adipose tissue depot may lead to greater understanding of obesity-associated cardiovascular diseases. Our study showed that pericardial adipose tissue is comprised primarily of Type 1 and Type 2 adipocytes, and understanding the role of these subpopulations in pericardial fat may be an interesting avenue of future research [10]. 

Recent studies utilizing high throughput single-cell RNA sequencing are also beginning to uncover a wide variety of heterogeneity both in gene expression and function of adipocyte precursor cells. These studies show that different populations of preadipocyte differ in beiging capacity, adipogenic differentiation, have differential wound healing and tissue remodeling capacities, and can differentially contribute to vasculogenesis [99,100,101,102]. Furthermore, single-cell RNA sequencing has been recently utilized to discover new adipocyte subpopulations. These include a subpopulation marked by CD142 expression that has the ability to inhibit adipocyte formation termed Adipocyte regulators cells (Areg) [100]. Aregs regulate adipocyte formation in a paracrine manner, are detected at a higher proportion in VAT compared to SAT, and are increased with obesity [100]. Another recently discovered subpopulation is marked by the expression of fibroblast-specific protein-1 (FSP1+), and is essential for the maintenance of the preadipocyte pool and its adipogenic potential [103]. Many of these subpopulations of preadipocytes are altered during the progression of diabetes and obesity, suggesting a possible role for these subpopulations in obesity complications.

Interestingly, the most commonly studied VAT, pgWAT, is an example of depot-specific function based on lineage heterogeneity. Studies in the late 1980s and early 1990s demonstrated that prWAT contains higher percentage of rapidly dividing preadipocytes than pgWAT, suggesting that prWAT preadipocytes are developmentally younger than that of pgWAT [42,104,105,106]. This phenomenon was found bilaterally and was independent of animal age, indicating the distinction was of embryonic origin [105,106]. Interestingly, recent studies on acylglycerophosphate acyltransferase family (AGPAT), enzymes in the formation of triacylglycerol precursor phosphatidic acid, clearly unmasked the heterogeneity of pgWAT. AGPAT4 deficiency leads to a highly sex and depot-specific weight gain only in the pgWAT in male mice, and not in female mice or other adipose tissue depots [107,108]. Compared to prWAT and other depots, pgWAT has been shown to be composed of adipocytes derived from progenitor cells from multiple lineages: Meox1/Pax3/Myf5, HoxB6, Wt1, Tagln, and bone marrow derived progenitors [10,11,64,82]. Sebo et al. showed that male pgWAT was derived from Meox1/Pax3 lineages, whereas female pgWAT was from HoxB6 lineage [82]. The mesothelium, smooth muscle/pericytes, and bone marrow, from which Wt1, Tagln, and bone marrow derived progenitors arise, respectively, are derivatives of lateral plate mesoderm. Therefore, adipocytes in male pgWAT are from both paraxial (Meox1/Pax3) and lateral plate mesoderm (Wt1/Tagln/bone marrow), while the majority of adipocytes in female pgWAT (HoxB6/Wt1/Tagln/bone marrow) are derived from the lateral plate mesoderm. The differences in the origin of these depots could potentially contributes to depot and sex dependent differences observed upon ablation of AGPAT4, and may also play a role in differences in adipose tissue deposition found in males and females.

## 6. Lipodystrophy 

Abnormal distribution of adipose tissues, as seen in lipodystrophy, supports the idea of developmentally distinct adipose tissue depots. Lipodystrophy is a collective disorder that can be classified as familial or acquired. While congenital generalized lipodystrophies (subdivision of familial lipodystrophy) are characterized by near complete loss of fat at early childhood excluding the mechanical adipose depots [109], familial partial lipodystrophies (FPL) include loss and gain of fat in various regions. One of the most common forms of FPL is FPL2 (also known as Dunnigan variety), which is characterized by the loss of subcutaneous fat in the extremities and trunk, but an accrual of fat in the visceral, head, and neck regions [110,111]. The excess accumulation of fat in the cervical region gives the appearance of buffalo-hump similar to Cushing’s syndrome. Although FPL forms are inheritable autosomal-dominant disorder involving mutations of vital adipocyte genes such as PPARG, PLN1, CIDEC, and AKT2 [112,113,114,115,116], the molecular mechanisms that selectively target adipocytes leading to loss or gain of fat in a particular region are currently unknown. Similarly, other forms of human WAT dystrophies, such as Barraquer–Simons syndrome, are characterized by facial fat loss and variable fat gain in the lower extremities suggests that this syndrome differentially affects cephalic SAT adipocytes derived from neural crest preadipocytes compared to mesodermal-derived adipocytes [117,118]. The phenotypes observed in Barraquer–Simons syndrome also suggests different origins for subcutaneous adipocytes in the lower versus upper body, and may also reflect different origins between deep and superficial SAT [11,57]. Thus, the selective loss and gain of fat in partial lipodystrophies, may, at least in part, be attributed to the heterogeneous nature of adipose tissues. 

Our study showed that Mx1-Cre and Wt1-Cre-derived adipocytes are mostly distributed in the scapular and visceral depots, respectively. The increase of visceral, head, and neck regions in FPL2 potentially suggest a selective accumulation in these two subpopulations. Furthermore, acquired lipodystrophy such as seen in patient treated with protease inhibitors in highly active antiretroviral therapy (HAART) for HIV may also be related to the heterogeneity of adipose tissues. The HAART treatment decreases transcription factors involved in adipocyte differentiation and function, interferes with fat storage, and increases adipocyte apoptosis, leading to adipose tissue dysfunction [119]. These patients generally have regional lipoatrophy of facial adipocytes, development of a “buffalo hump” of massively increased dorsocervical adipose tissue, increase visceral fat, and elevating the risk of metabolic disease [120,121]. Studies in mice show numerous adipose subpopulations that are highly concentrated in a dorsoanterior manner, including the Meox1/Pax3, 7/Myf5 and Mx1-derived adipocytes. Thus, we hypothesize HAART may differentially affect some adipocyte subpopulations.

## 7. Conclusions

The obesity epidemic is a severe public health crisis. However, in recent years, great progress has been made in elucidating the contribution of white, brite, and brown adipose tissues to metabolism and whole body physiology. In addition, recent studies have begun to examine the developmental and functional heterogeneity found within the WAT, BAT, and brite adipocyte populations. This cellular heterogeneity is an emerging concept, and the physiological significance of this heterogeneity is just beginning to be explored. In this review we have touched upon numerous issues, including the metabolic contribution of different white adipose tissue depots, adipose tissue deposition, and partial lipodystrophies, in which these adipocyte subpopulations may play critical roles. These avenues of research hold promise to developing targeted interventions to combat the comorbidities associated with obesity.

## Figures and Tables

**Figure 1 biology-08-00023-f001:**
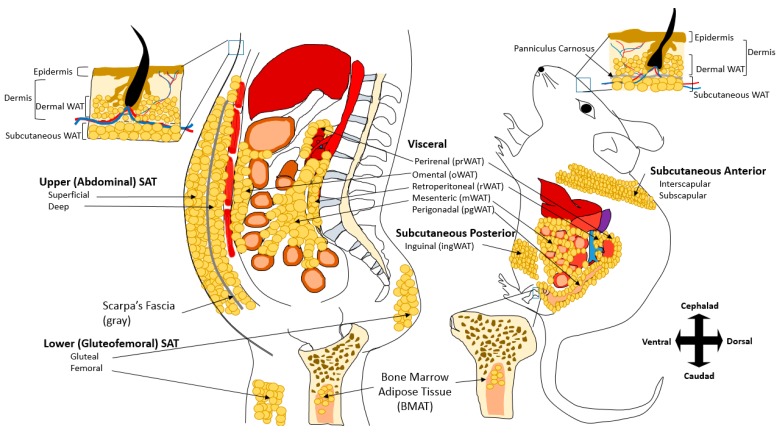
Comparison between Human and Rodent Adipose Tissues. White Adipose Tissues (WAT) are mostly visceral adipose tissue (VAT) and subcutaneous adipose tissue (SAT). Major visceral depots common in both humans and rodents include epicardial/pericardial (not shown), perirenal (prWAT), retroperitoneal (rWAT), and mesenteric WAT (mWAT). While humans have large omental (oWAT) fat, mice have large perigonadal (pgWAT) fat. SAT in humans can be divided into upper (abdominal) and lower (gluteofemoral). Abdominal fat in human can be further divided into superficial and deep, which are separated anatomically by Scarpa’s Fascia. In mice, SAT is divided into anterior and posterior. Dermal WAT (dWAT) is another fat depot that exists in both humans and mice. dWAT is located in the dermis and above the SAT (note: this separation is visible by the presence of striated muscular layer known as panniculus carnosus only found in mice). Bone marrow adipose tissue (BMAT or MAT) is also another adipose depot common in both species. Cephalad and caudad are also referred to as anterior and posterior in mice, respectively.

**Figure 2 biology-08-00023-f002:**
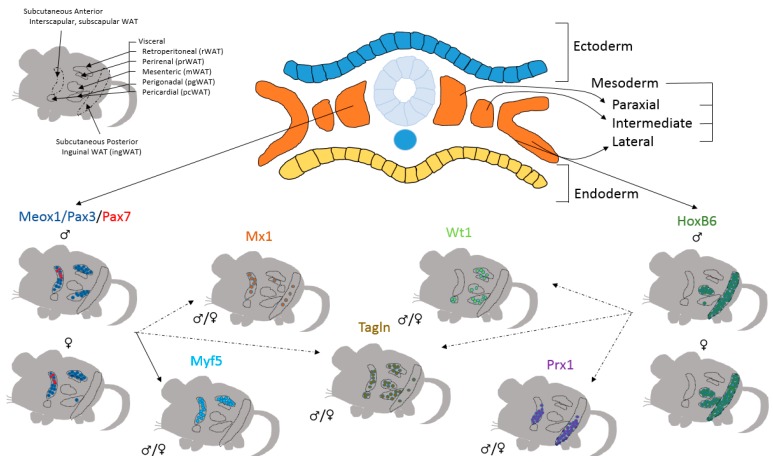
Heterogeneous Origins of White Adipose Tissue Depots. Adipose precursor cells of individual adipose depots originate from distinct areas of mesoderm. While Meox1/Pax3/Pax7- (from paraxial mesoderm) derived adipocytes predominate in the dorsoanterior depots, HoxB6- (lateral plate mesoderm) derived adipocytes are found largely in the ventral regions. Like Meox1/Pax3/Pax7 derived adipocytes, Mx1 and Myf5 derived adipocytes are also found in dorsoanterior depots. Adipocytes derived from the Prx1 lineage are found only subcutaneous depots, while adipocytes derived from the mesothelium and marked by Wt1 expression are only found in the visceral depots. Adipocytes derived from the Tagln lineage are found in all adipose tissue depots. Dotted lines indicate a potential connection among these lineages.

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
