# Peer review of "Deciphering White Adipose Tissue Heterogeneity"

_biology, 2019, doi:10.3390/biology8020023_

Reviewer 1 Report

Luong et al. have presented an important, comprehensive review discussing the heterogeneity of white adipose tissue.

Overall, the content presented is solid – the review in general is written a little disjointedly and contains several run on sentences that take away from the importance of the information being discussed. 

One thing that seems to be missing is how the heterogeneity of adipose tissue affects responsiveness to stimuli? Things such as cold, b3, exercise, and environment could all be important factors in responsiveness and adaptations to the different AT depots; is there any information on the heterogeneity of those depots that could indicate how likely they would be to be responsive to different stimuli?  I think that would be very important.

Line 62 – re-phrase the introductory sentence – begins by stating there are two types of AT and then lists several others.

Line 63 – don’t need the word “fat” after SAT.

Line 110 – reference to ‘our study’ should be expanded so the reader knows what is being discussed. To this point, only metabolic differences in SAT and VAT have been mentioned.

Line 134 – is this referring to all adipose tissue depots or one in particular?  Since the previous section the different metabolic effects of these depots, it needs to be clarified.

Line 138 – explain the FACS study (Rodeheffer 2008) in more detail; it is mentioned as a key contributor to defining cell populations and advancing the field, but not discussed. 

Line 152-179 – The data presented here covers very important information but reads very disjointedly and presents such different information in a brief two paragraph that it is hard to follow/interpret. I suggest breaking down these paragraphs into different subsections (each discussing the lineage) for clarity – otherwise that gets lost in the text.

Line 235 – I had anticipated this section (Functional Heterogeneity of White Adipose Tissue) to be the crux of the review and was surprised to find it was only one paragraph.  I think this information could be expanded even to discuss differences in gene expression in this AT depots (Gesta and Kahn 2006, 2007) to expand upon these data and provide an important perspective on the existing literature. The senior author has extensive knowledge of these data.

Line 249 – The discussion of lipodystrophies is a unique angle as most reviews discussing differences in adipose tissue would discuss obesity.  However, it seems out of place here based on the previous sections discussed.  Suggest adding some contrast, discussing obesity and lipodystrophies to put in context and provide a better overview of how the heterogeneity of adipose tissue could contribute to different disease states.

Author Response

We would like to thank the reviewers for their careful reading and useful suggestions. As suggested by the reviewers, we have expanded our discussion on the functional heterogeneity and influence of external factors to adipocyte subpopulations.  We have also included the insightful information provided by single-cell RNAseq, clarified the text, and proofread for grammatical errors. The specific and additional changes are outlined below.

REVIEWER1

Luong et al. have presented an important, comprehensive review discussing the heterogeneity of white adipose tissue. 

Overall, the content presented is solid – the review in general is written a little disjointedly and contains several run on sentences that take away from the importance of the information being discussed.

We apologize for the grammatical issues. We have reorganized and proofread the manuscript before resubmission.

One thing that seems to be missing is how the heterogeneity of adipose tissue affects responsiveness to stimuli? Things such as cold, b3, exercise, and environment could all be important factors in responsiveness and adaptations to the different AT depots; is there any information on the heterogeneity of those depots that could indicate how likely they would be to be responsive to different stimuli?  I think that would be very important.

We would like to thank the reviewer for these suggestions. We agree that these are important topics to be discussed in the review. Although there is not a great deal of information about how heterogeneity of white adipocytes regulates whole body physiology, there are some emerging studies that indicate that adipocyte heterogeneity may regulate the response of adipose tissue to adrenergic stimulation, obesity, and sex-dependent fat deposition.  Many of these topics are now discussed in the “Functional Heterogeneity” section of the manuscript (Lines 234-292).

Line 62 – re-phrase the introductory sentence – begins by stating there are two types of AT and then lists several others. 

As suggested, the introductory sentence has been changed to “In general, the majority of WAT is categorized as either subcutaneous (SAT) and visceral (VAT) adipose tissue, with the exception of smaller adipose depots including the dermal WAT (dWAT) and bone marrow adipose tissue (MAT) that are distinct from SAT and VAT.”

Line 63 – don’t need the word “fat” after SAT. 

We apologize for the typos. The word has been deleted as suggested.

Line 110 – reference to ‘our study’ should be expanded so the reader knows what is being discussed. To this point, only metabolic differences in SAT and VAT have been mentioned. 

We agree that the reference to ‘our study’ was not clearly explained. A much more thorough explanation of the study has been given in the second paragraph of the introduction (Lines 36-45).

Line 134 – is this referring to all adipose tissue depots or one in particular?  Since the previous section the different metabolic effects of these depots, it needs to be clarified. 

We would like to thank the reviewer for pointing this out. This section refers to the preadipocyte lineage of all adipose tissue depots, and has been clarified in the text.

Line 138 – explain the FACS study (Rodeheffer 2008) in more detail; it is mentioned as a key contributor to defining cell populations and advancing the field, but not discussed.  

We agree with the reviewer that this study by Rodeheffer et al. 2008 represents a seminal advance. We have included a brief discussion of the key findings of this manuscript (Lines 140-151).

Line 152-179 – The data presented here covers very important information but reads very disjointedly and presents such different information in a brief two paragraph that it is hard to follow/interpret. I suggest breaking down these paragraphs into different subsections (each discussing the lineage) for clarity – otherwise that gets lost in the text. 

We agree with the reviewer and have attempted to make this section of the manuscript much more readable.  In this section of the text, we now describe each developmental lineage in its own separate paragraph for clarity.

Line 235 – I had anticipated this section (Functional Heterogeneity of White Adipose Tissue) to be the crux of the review and was surprised to find it was only one paragraph.  I think this information could be expanded even to discuss differences in gene expression in this AT depots (Gesta and Kahn 2006, 2007) to expand upon these data and provide an important perspective on the existing literature. The senior author has extensive knowledge of these data. 

We agree with the reviewer and have expanded the ‘Functional Heterogeneity of White Adipose Tissue’ section. We have added the details on how external stimuli influence the functional properties of the subpopulations based on currently known data. We have also discussed the functional properties that are potentially influenced by the development of each subpopulation (Lines 234-292).

Line 249 – The discussion of lipodystrophies is a unique angle as most reviews discussing differences in adipose tissue would discuss obesity.  However, it seems out of place here based on the previous sections discussed.  Suggest adding some contrast, discussing obesity and lipodystrophies to put in context and provide a better overview of how the heterogeneity of adipose tissue could contribute to different disease states.

We agree that obesity should be used for comparison to lipodystrophies, and added a clarifying paragraph as suggested. We have also further revised the manuscript to better reflect the possible influence of heterogeneity in adipose tissues on both of these pathological states.

Reviewer 2 Report

It has been long known that different adipose tissue depots have different characteristics and function. However, it is only recently that the field has started appreciating the large heterogeneity that occurs within a single depot, both at the preadipocyte and mature adipocyte level. In particular, the single cell RNAseq technology has provided a powerful tool to address adipose tissue heterogeneity.

The review by Luong et al. on “Deciphering White Adipocyte Heterogeneity” aiming at summarizing the current knowledge on the topic is timely. The review carefully describes the current view of white adipose precursor populations, and the potential implication for lipodystrophies. However, several omissions and concerns need to be addressed before publication:

The manuscript provides a brief summary of the different white adipose tissue depots in mice and humans. However, the pericardial/epicardial is not mentioned although it represents an important depot. Also, there are striking differences between mouse and human adipose tissue depots and those need to be described. Humans and mice sometimes display opposing gene expression patterns in different depots (PMID: 28529941)

Key publications have started to emerge, using single cell RNAseq technology to characterize adipose tissue heterogeneity (PMID: 29925944; PMID: 29937373 for example). These seminal papers are not cited, nor is discussed single cell RNAseq as a method to study adipose tissue heterogeneity

A novel FACS method which allows to study mature white adipocyte heterogeneity has recently been developed, and is also not mentioned in the review. It allowed to demonstrate that a subset of human adipocytes lacks the β2-adrenergic receptor (PMID: 30184507)

The text needs to be carefully checked for inaccuracy, incoherence, grammar error and typos. Some of the errors I noticed are listed below:

-          line 39: “brite/beige adipocytes interspersed within WAT that are capable of transforming into thermogenic BAT-like adipocytes”. Incoherent sentence. Brite/beige adipocytes are thermogenic BAT-like adipocytes

-          line 47: Refs need to be included

-          line 109: “gene-expression patterns after cell culture passages”. Change to “gene expression patterns after culture

-          line 121: what is meant by “low lipoprotein lipase”?

-          line 138: “In this study, …” which study are the authors referring to?

-          Line 150: replace “increased” by higher

-          Line 170: delete “and”

-          Line 175: “The number of these adipocytes” which adipocytes are the author referring to?

-          Line 177: “in contrast to these, a separate study” What is meant by in contrast to these?

-          Line 186: “factors crucial for the myogenic, also give rise” incoherent sentence

-          Line 188: “a downstream target gene of Pax3/7, also adipocytes in the interscapular WAT”. Incoherent sentence

-          Line 209: “While Mx1 and Prx1 derived adipocytes are only.” Incoherent

-          Line 250: “Abnormal distributions of adipose” Remove s

-          Line 261: “Similarly, other are forms”. What is meant by other are?

-          Line 268: “lipodystrophies, may, at least in part, can be attributed” incoherent

-          Line 274: “adipocyte differentiation and function, and interferes with fat storage, and increases…” Remove and before interferes

Author Response

We would like to thank the reviewers for their careful reading and useful suggestions. As suggested by the reviewers, we have expanded our discussion on the functional heterogeneity and influence of external factors to adipocyte subpopulations.  We have also included the insightful information provided by single-cell RNAseq, clarified the text, and proofread for grammatical errors. The specific and additional changes are outlined below.

REVIEWER2

It has been long known that different adipose tissue depots have different characteristics and function. However, it is only recently that the field has started appreciating the large heterogeneity that occurs within a single depot, both at the preadipocyte and mature adipocyte level. In particular, the single cell RNAseq technology has provided a powerful tool to address adipose tissue heterogeneity.

The review by Luong et al. on “Deciphering White Adipocyte Heterogeneity” aiming at summarizing the current knowledge on the topic is timely. The review carefully describes the current view of white adipose precursor populations, and the potential implication for lipodystrophies. However, several omissions and concerns need to be addressed before publication.

The manuscript provides a brief summary of the different white adipose tissue depots in mice and humans. However, the pericardial/epicardial is not mentioned although it represents an important depot. Also, there are striking differences between mouse and human adipose tissue depots and those need to be described. Humans and mice sometimes display opposing gene expression patterns in different depots (PMID: 28529941)

We would like to thank the reviewer for the suggestions and the references. We have added further information on the epicardial/pericardial depot, as suggested (Lines 247-256). In addition, we have also discussed the differential gene expression between mice and humans, especially noted in studies of markers of browning (Lines 123-128). 

Key publications have started to emerge, using single cell RNAseq technology to characterize adipose tissue heterogeneity (PMID: 29925944; PMID: 29937373 for example). These seminal papers are not cited, nor is discussed single cell RNAseq as a method to study adipose tissue heterogeneity.

We agree that single-cell RNAseq is providing insightful information about adipose heterogeneity. As suggested, we have revised the manuscript to reflect the contribution of these recent single-cell RNAseq publications on understanding adipocyte subpopulations (Lines 256-270).

A novel FACS method which allows to study mature white adipocyte heterogeneity has recently been developed, and is also not mentioned in the review. It allowed to demonstrate that a subset of human adipocytes lacks the β2-adrenergic receptor (PMID: 30184507)

We apologize for the oversight and have now added this interesting study to the manuscript (Ref 89).

The text needs to be carefully checked for inaccuracy, incoherence, grammar error and typos. Some of the errors I noticed are listed below:

We apologize for many grammatical errors and typos.  In all cases, we have revised the manuscript as suggested and proofread for grammatical errors.

-          line 39: “brite/beige adipocytes interspersed within WAT that are capable of transforming into thermogenic BAT-like adipocytes”. Incoherent sentence. Brite/beige adipocytes are thermogenic BAT-like adipocytes

-          line 47: Refs need to be included

-          line 109: “gene-expression patterns after cell culture passages”. Change to “gene expression patterns after culture

-          line 121: what is meant by “low lipoprotein lipase”?

-          line 138: “In this study, …” which study are the authors referring to?

-          Line 150: replace “increased” by higher

-          Line 170: delete “and”

-          Line 175: “The number of these adipocytes” which adipocytes are the author referring to?

-          Line 177: “in contrast to these, a separate study” What is meant by in contrast to these?

-          Line 186: “factors crucial for the myogenic, also give rise” incoherent sentence

-          Line 188: “a downstream target gene of Pax3/7, also adipocytes in the interscapular WAT”. Incoherent sentence

-          Line 209: “While Mx1 and Prx1 derived adipocytes are only.” Incoherent

-          Line 250: “Abnormal distributions of adipose” Remove s

-          Line 261: “Similarly, other are forms”. What is meant by other are?

-          Line 268: “lipodystrophies, may, at least in part, can be attributed” incoherent

-          Line 274: “adipocyte differentiation and function, and interferes with fat storage, and increases…” Remove and before interferes

Round  2

Reviewer 2 Report

The changes that the authors made to the manuscript significantly improved its quality.

The authors added a paragraph on epicardial fat in the section “Functional Heterogeneity of Adipocytes”. However, epicardial fat is not mentioned in the first section describing the different “Adipose Tissue Depots”. That is missing

Line 242: “Previously, we demonstrated that Tbx15 expression, which is highly expressed in SAT preadipocytes and adipocytes, is due to the presence of subset of these cells.” Unclear what is meant here. Please rephrase

Author Response

The changes that the authors made to the manuscript significantly improved its quality.

We again would like to thank the reviewer for the helpful comments in insights which have improved this manuscript.

The authors added a paragraph on epicardial fat in the section “Functional Heterogeneity of Adipocytes”. However, epicardial fat is not mentioned in the first section describing the different “Adipose Tissue Depots”. That is missing

As suggested by the reviewer, we clarified the definition of visceral fat, which now includes the epicardial, and other depots in the section “Adipose Tissue Depots” (line 62-63).

Line 242: “Previously, we demonstrated that Tbx15 expression, which is highly expressed in SAT preadipocytes and adipocytes, is due to the presence of subset of these cells.” what is meant here. Please rephrase

We apologize for the awkward sentence. We have edited the sentence to refer to the presence of Tbx15-expressing adipocytes.